# Codon Usage Provides Insights into the Adaptive Evolution of Mycoviruses in Their Associated Fungi Host

**DOI:** 10.3390/ijms23137441

**Published:** 2022-07-04

**Authors:** Qianqian Wang, Xueliang Lyu, Jiasen Cheng, Yanping Fu, Yang Lin, Assane Hamidou Abdoulaye, Daohong Jiang, Jiatao Xie

**Affiliations:** 1State Key Laboratory of Agricultural Microbiology, Huazhong Agricultural University, Wuhan 430070, China; qianqian_wang@mail.hzau.edu.cn (Q.W.); lvxueliang0715@webmail.hzau.edu.cn (X.L.); jiasencheng@mail.hzau.edu.cn (J.C.); doul_as@yahoo.fr (A.H.A.); daohongjiang@mail.hzau.edu.cn (D.J.); 2The Hubei Key Lab of Plant Pathology, College of Plant Science and Technology, Huazhong Agricultural University, Wuhan 430070, China; yanpingfu@mail.hzau.edu.cn (Y.F.); yanglin@mail.hzau.edu.cn (Y.L.); 3Hubei Hongshan Laboratory, Wuhan 430070, China

**Keywords:** mycovirus, codon usage bias, natural selection, transcription, RNA biosynthetic process

## Abstract

Codon usage bias (CUB) could reflect co-evolutionary changes between viruses and hosts in contrast to plant and animal viruses, and the systematic analysis of codon usage among the mycoviruses that infect plant pathogenic fungi is limited. We performed an extensive analysis of codon usage patterns among 98 characterized RNA mycoviruses from eight phytopathogenic fungi. The GC and GC3s contents of mycoviruses have a wide variation from 29.35% to 64.62% and 24.32% to 97.13%, respectively. Mycoviral CUB is weak, and natural selection plays a major role in the formation of mycoviral codon usage pattern. In this study, we demonstrated that the codon usage of mycoviruses is similar to that of some host genes, especially those involved in RNA biosynthetic process and transcription, suggesting that CUB is a potential evolutionary mechanism that mycoviruses adapt to in their hosts.

## 1. Introduction

Within the standard hereditary code, 20 amino acids are encoded by 61 sense codons, and 3 (TAG, TAA, and TGA) of them are termination codons. With the exception that two amino acids (Met and Trp) are encoded by a single codon, the other eighteen amino acids are encoded by two or more codons, resulting in codon redundancy. Diverse codons encode the same amino acid, which is known as synonymous codons. Synonymous codons are not utilized with same frequency during the process of transcription and translation. A few synonymous codons are utilized more regularly than others, which leads to a phenomenon that is called codon usage bias (CUB) [1,2].

CUB plays crucial roles in protein function, translation efficiency, and/or accuracy [3,4]. CUB has facilitated the understanding of genome evolution [5], heterologous gene expression [6], gene-level prediction, and gene function [7,8]. Multiple factors could be involved in the occurrence course of CUB, and two of them, mutation pressure and natural selection, are fundamental driving forces for CUB [9,10]. Notably, there are other factors associated with CUB, including base composition [11], gene expression level and length [12,13], protein secondary structure, protein hydrophobicity and hydrophilicity [14], and tRNA abundance and type [15,16].

CUB commonly occurs in viruses, and systematic CUB analysis has been conducted in some essential viruses, from plant viruses [17,18], animal viruses [19], and insect viruses [20] to human viruses [21,22]. Viruses are pure parasites and rely on the host cellular machinery for replication. The CUB of viruses provides another example to explain how viruses adapt to expression conditions in their hosts. Viral coding sequences, especially those with high expression, have similar CUB to their hosts [23]. Thus, CUB analysis provides an alternate perspective to understand viral molecular evolution and ecology better. The codon usage pattern can reflect some changes in virus evolution, allowing viruses to optimize their survival and improve their fitness in their hosts [24,25,26], especially when escaping the host immune system [27]. Co-evolution between viruses and susceptible hosts (plant or animal) has been extensively elaborated at the codon usage level [18,19,28].

Mycoviruses are prevalent in all major taxonomic groups of fungi [29]. All known mycoviruses have diverse genomic types, including double-stranded RNA (dsRNA), positive-sense single-stranded RNA ((+)ssRNA), negative-sense single-stranded RNA ((−)ssRNA), and single-stranded circular DNA. An expanding number of novel mycoviruses have been unremittingly reported due to the broad application of high-throughput sequencing technologies. Most known mycoviruses with latent infection are adapted to live within their hosts for long periods or have unknown functions in their hosts [30,31]. Some studies of the codon usage among mycoviruses have been reported. Je et al. (2019) analyzed the codon usage pattern of the RNA-dependent RNA polymerase (RdRP) genes of the *Aspergillus*-infecting viruses, AfuPmV-1 has the stronger codon usage bias relative to other viruses, and AfuPmV-1 has higher adaptability to human and fungal hosts compared with other [32]; the sequence characteristics and codon usage patterns of RdRP and coat protein (CP) genes among *Partitiviridae* viruses that infect fungi were analyzed [33]. Simón et al. (2021) studied the relation between viruses and the respective hosts by analyzing the GC content of viruses and hosts, including mycoviruses [34]. Although the discovery and research on mycoviruses have mainly focused on plant pathogenic fungi, the systematic analysis of codon usage among mycoviruses infected plant pathogenic fungi is limited.

In this study, we retrieved 98 RNA mycoviral sequences from the National Center for Biotechnology Information (NCBI) database, including dsRNA, (+)ssRNA, (−)ssRNA mycoviruses. We further evaluated the CUB of the mycoviruses studied here based on replication-associated polyprotein (RP) coding sequences. We also investigated the main factors affecting CUB in mycoviruses and the relationship of codon usage between mycoviruses and fungal genes. Our study could fill a knowledge gap in understanding the mycovirus codon usage pattern and presents intriguing possibilities to study mycoviruses-fungi co-evolution.

## 2. Results

### 2.1. Analysis of Mycoviral RP Genes Nucleotide Composition

The nucleotide composition could profoundly influence the CUB of a coding sequence [11,34]. Therefore, to understand the potential influence of compositional constraints on mycoviral codon usage, the nucleotide composition of mycoviral RP-coding sequences was analyzed (including A%, C%, T%, G%, and GC%). The results showed an unequal distribution of A-, C-, T-, and G- among the mycoviral RP-coding sequences (Appendix A). The overall GC contents range widely from 29.35% (Botrytis cinerea negative-stranded RNA virus 1, BcNSRV1) to 64.62% (Rhizoctonia solani fusarivirus 1, RsFV1) among the mycoviruses. Eighty-eight mycoviruses have 40–60% GC values, while ten mycoviruses are outside 35–60% of the interval. The mean GC% of (−)ssRNA, (+)ssRNA, and dsRNA mycoviruses includes 37.76% ± 5.52%, 49.71% ± 6.68%, and 50.44% ± 5.30%, respectively, and (+)ssRNA and dsRNA mycoviruses have a higher GC content than (−)ssRNA mycoviruses (Figure 1A). The GC content’s comparison within different taxonomic families showed that the GC contents of members within *Endornaviridae* and *Partitiviridae* families are 40.76% ± 3.44% and 46.51% ± 3.28%, respectively, and those within *Totiviridae*, *Tymovirales*, and *Botourmiaviridae* exceeded more than 50% (Figure 1B).

In addition to the nucleotide composition, GC3s (the frequency of the third nucleotides G + C in the synonymous codons) contents of mycoviruses were also investigated. Various mycoviruses have their own A/T-ended and C/G-ended codon preference due to compositional constraints (Appendix A). It is different from mycoviral GC contents analysis, the extreme GC3s values occurred within Rhizoctonia solani bunya/phlebo-like virus 1 (RsBPLV1, 97.13%) and BcNSRV1 (24.32%) (Appendix A). Of codons related to the entire 18 amino acids (except Met and Trp), 23 over-represented codons (the relative synonymous codon usage (RSCU) ≥ 1.6) in RsBPLV1 all end with C or G, and 30 of the 32 under-represented codons (RSCU ≤ 0.6) in RsBPLV1 all end with A or T (Appendix A). Contrarily to RsBPLV1, A- or T-ended codons were strongly preferred in BcNSRV1, and it does not prefer to use C- and G-ended codons (Appendix A). The trend of GC3s content analysis was similar to that of GC% content analysis in mycoviruses (Figure 1C,D). (+)ssRNA and dsRNA mycoviruses have a higher GC3s content than (−)ssRNA mycoviruses, and *Endornaviridae* mycoviruses have the lowest GC3s values (37.87% ± 7.68%) compared with other families. This unequal nucleotide usage suggested that GC and GC3s could contribute in influencing mycoviral CUB.

### 2.2. Overall CUB among Mycoviral RP Genes

To measure the extent of mycoviral CUB, the effective number of codon (ENc) values were estimated for mycoviral RP genes. Of the 98 mycoviruses, RsBPLV1 displays ENc value < 35 (28.28), indicating that RsBPLV1 has a dramatic codon usage bias. ENc values of twenty-three mycoviruses range from 40 to 50, and seventy range from 50 to 60. Furthermore, ENc values of three mycoviruses, including Rosellinia necatrix hypovirus 2 (RnHV2), Magnaporthe oryzae ourmia–like virus 4, and Sclerotinia sclerotiorum ourmia-like virus 4, are more than 60 (Figure 2, Appendix A). Those results revealed that the CUB among mycovirus is weak. The ENc values of (−)ssRNA mycoviruses (48.88 ± 5.53) are lower than those of (+)ssRNA and dsRNA mycoviruses (53.85 ± 4.39 and 53.10 ± 3.54, respectively). ENc values (56.35 ± 1.85) of members within *Hypoviridae* are the highest at the family level.

### 2.3. The Contribution of Mutation Pressure and Natural Selection in Forming Mycoviral CUB

To determine the potential roles of mutation pressure and natural selection in shaping the CUB of mycoviral RP-coding sequences, we performed ENc-GC3s, Parity Rule 2 (PR2), and neutrality plot analyses.

ENc-GC3s analysis. The ENc value of each mycoviral RP gene was plotted against the corresponding GC3s content (Figure 3A). In the case of the true mycoviral ENc values are lying on or just under the curve of the expected ENc values, it means that the CUB is mainly constrained by mutation pressure; otherwise, natural selection plays non-negligible roles in shaping the mycoviral CUB [35]. Ninty-five of the ninty-eight mycoviruses were below the blue theoretical curve (Figure 3A). Furthermore, person correlation analysis revealed a weak negative correlation between ENc and GC3s (r = −0.23, *p* < 0.05). These results suggesting that mycoviral CUB is shaped by both mutational pressure and natural selection.

PR2 analysis. G and C (or A and T) are proportionally used among the degenerate codon groups in one gene when CUB is affected by the mutational pressure alone [36]. Natural selection would not necessarily cause the proportionate use of nucleobase (G and C or A and T). The PR2 plot is a measure of intra-strand bias, which demonstrates the influence of mutation and selection stress on the codon usage of genes. In a PR2 plot, in the case where mycoviruses are lying on or near the midpoint, this means that CUB is constrained by mutation pressure, otherwise, natural selection is the main force in shaping the mycoviral CUB [36]. In our analysis, we set the values of A3/(A3 + T3)|4 (the third codon position of the four-codon amino acids) and G3/(G3 + C3)|4 as the ordinate and abscissa, respectively. As shown in Figure 3B, all the points were away from the origin point (0.5, 0.5), suggesting that mutation pressure and natural selection all can affect mycoviral CUB.

Neutrality plot analysis. Previously, the ENc-GC3s and PR2 plot indicated that mycoviral CUB is influenced by natural selection and mutational pressure. Then, we performed the neutrality plot analysis to quantify the relative contributions of mutation pressure and natural selection on mycoviral CUB, when the regression coefficient is near 1, CUB is mainly affected by mutation pressure and is the dominant factor, in contrast, if the regression coefficient is near 0, natural selection plays the key role. Considering that we selected the mycoviruses from eight different hosts, we conducted the neutrality plot analysis (GC3 (GC values at the third position of codons) was plotted against GC12 (the average of GC values at the first (GC1) and second (GC2) positions of codons)) of mycoviruses that infected the same fungi host, and the outlier GC12 or GC3 values among mycoviruses that infected the same host were removed. As shown in Figure 4, the slopes of the regression lines in *C. parasitica*, *R. necatrix*, *B. cinerea*, *F. graminearum*, *R. solani*, *M. oryzae*, *U. virens*, and *S. sclerotiorum* were 0.0978, 0.4801, 0.1794, 0.3485, 0.4708, 0.2088, 0.2577, and 0.3497, respectively. All slopes were less than 0.5; this implied that mycoviral codon usage is mainly influenced by natural selection.

### 2.4. The Relative Abundance of Dinucleotide among Mycoviral RP Genes

Previous studies have suggested that the relative dinucleotide abundance can also affect the codon usage of virus [37]. The relative abundances of 16 dinucleotides from the mycoviral RP sequences were calculated to investigate the dinucleotides effect on codon usage in mycoviruses. All dinucleotides were non-randomly distributed (Figure 5 and Appendix A). The frequencies of CpA, GpA, and TpG were the three highest of the 16 dinucleotides, and the mean values were 1.11 ± 0.15, 1.17 ± 0.15, and 1.18 ± 0.14, respectively. TpA had the lowest mean value (0.69 ± 0.14) and was under-represented (*ρ_xy_* < 0.78), especially in members of *Hypoviridae* (0.62 ± 0.05, except RnHV2), *Partitiviridae* (0.66 ± 0.08), *Tymovirales* (0.55 ± 0.10), and *Botourmiaviridae* (0.68 ± 0.07, except Magnaporthe oryzae ourmia-like virus 1). The RSCU values analysis of six codons containing TpA (TTA, CTA, ATA, GTA, TAT, and TAC) suggested that these codons were not preferentially used (Appendix A). Moreover, there were several over-represented dinucleotides (*ρ_xy_* > 1.23) (Appendix A, marked with red), and mycoviruses preferred to use the codons containing these over-represented dinucleotides (Appendix A). Overall, these findings revealed that dinucleotide composition plays potential roles in mycoviral CUB.

### 2.5. The Similarity of Codon Usage between Mycoviruses and Their Fungi Host

The codon usage pattern of viruses is also affected by the host [24,38]. The four phytopathogenic fungi, *Rhizoctonia solani*, *Fusarium graminearum*, *Sclerotinia sclerotiorum*, and *Botrytis cinerea*, have well genomic annotations. Moreover, mycoviruses in these four fungi showed rich diversity and are abundant in number [39,40,41,42]. To determine the similarity for codon usage patterns between mycoviruses and fungal genes, we performed principal component analysis (PCA) based on the RSCU values of mycoviruses and the four fungal genes (Appendix A). The values of the first four axes of RSCU-PCA for *R.solani* genes and mycoviruses were 11.88%, 5.24%, 5.12%, and 3.06%; those observed for *F. graminearum* genes and mycoviruses were 22.70%, 4.31%, 3.07%, and 2.93%; for *S. sclerotiorum* genes and mycoviruses, they were 13.07%, 7.55%, 5.74%, and 4.71%, while for *B. cinerea* genes and mycoviruses, they were 13.99%, 7.79%, 4.92%, 4.06%. These results revealed that axis1 was the major factor in determining codon usage pattern for the mycoviruses and fungal genes. Then, axis 1 (PC1) was plotted against axis 2 (PC2) (Figure 6). The mycoviruses showed a scattered distribution, and the observations verified that there were differences in the codon usage pattern among the mycoviruses that infect the same host. Some mycoviruses, such as Rhizoctonia solani negative-stranded virus 2, Fusarium graminearum mycotymovirus 1 (FgMTV1), Sclerotinia sclerotiorum RNA virus L (SsRV-L), and Botrytis virus X (BVX), have higher PC1 or PC2, indicating that the codon usage patterns of these mycoviruses were highly different from that of fungal genes and other mycoviruses; however, there were 11, 5, 9, and 5 mycoviruses that were relatively concentrated together in *R. solani*, *F. graminearum*, *S. sclerotiorum*, and *B. cinerea*, respectively (Figure 6 and Appendix A). The average PC1 and PC2 of these mycoviruses were calculated, and then the average PC1 and PC2 were set as the center of a circle with the radius calculated to be minimal. To cover those mycoviruses, the fungal genes that were in the circles were taken as host genes that have similar codon usage patterns to mycoviruses (Appendix A).

To investigate the contributions of fungal genes that have similar codon usage pattern to mycoviruses in the biological processes of fungi, we analyzed these fungal genes for enriched to gene ontology (GO) terms in biological processes. These fungal genes are enriched 15, 44, 78, and 21 GO terms in *R. solani*, *F. graminearum*, *S. sclerotiorum*, and *B. cinerea*, respectively (Appendix A). As shown in Figure 7, the fungal genes that have similar codon usage patterns to mycoviruses were enriched, including the top 20 of GO terms in the biological processes of *R. solani* (Figure 7A), *F. graminearum* (Figure 7B), *S. sclerotiorum* (Figure 7C), and *B. cinerea* (Figure 7D). The GO enrichment results of *F. graminearum*, *S. sclerotiorum*, and *B. cinerea* shared 10 identical GO terms (Figure 7B−D) involved in nucleobase-containing compound metabolic/biosynthetic process, heterocycle biosynthetic process, RNA biosynthetic process, and transcription, and the results of the four hosts (*R. solani*, *F. graminearum*, *S. sclerotiorum*, and *B. cinerea*) shared four identical GO terms involved in transcription (Figure 7).

Additionally, with the exception of the 11, 5, 9, and 5 mycoviruses in *R. solani*, *F. graminearum*, *S. sclerotiorum*, and *B. cinerea* that have been analyzed above (listed in Appendix A), we also analyzed the function of the fungal genes that have similar codon usage pattern to other mycoviruses. The PC1 and PC2 of each remaining mycovirus were set as the circle’s center with a radius of 0.3 units, and the fungal genes that are located within the circle were taken as the host genes with RSCUs similar to this mycovirus. We only selected the two mycoviruses with the least number of similar host genes for analysis in each fungus (Appendix A). While these mycoviruses were visibly distant from the majority of fungal genes and they were close to fungal genes with different function, the fungal genes involved in transcription also can be found around RsFV1 and BcNSRV1. The host genes with similar codon usage to two *Tymovirales* mycoviruses, FgMTV1 and Sclerotinia sclerotiorum mycotymovirus 1, were involved in protein modification, transport, ubiquitination and heterodimerization (Appendix A).

Most of viruses do not have their own tRNAs, and the translation of viral proteins depends entirely on the host’s tRNA pool [26]. We received the copy number of tRNAs for *R. solani*, *F. graminearum*, *S. sclerotiorum*, and *B. cinerea* from GtRNAdb. The tRNAs corresponding to some codons are missing from the four fungal genomes. We analyzed the use of these missing tRNA-corresponding codons among mycoviruses that infected the four fungi (Appendix A). The RSCU values of these codons among most mycoviruses and the four fungi fell within the normal frequency range (0.6 ≤ RSCU ≤ 1.6). However, in the mycoviruses that were visibly distant from the majority of fungal genes, these codons were over-represented than other mycoviruses, such as SsRV-L, which had seven over-represented codons, and there were 8, 8, and 6 codons in BVX, FgMTV1, and RsBPLV1, respectively (Appendix A). These resulted suggesting that the compatibility of mycovirus synonymous codon usage and host tRNA abundance can influence the similarity between virus and host on codon usage.

### 2.6. The Expression of Fungal Genes That Have Similar Codon Usage Pattern to Mycoviruses Response to Mycoviruses Infections

A certain number of fungal genes could respond to mycovirus infection, whether upregulated or downregulated [43,44,45]. However, it is unclear how the fungal genes that have similar codon usage pattern to mycoviruses are changed upon mycoviruses infections. To answer this question, we detected the transcriptional changes in *S. sclerotiorum* following the infection of Sclerotinia sclerotiorum mycoreovirus 4 (SsMYRV4) using RNA-Seq. The upregulated and downregulated *S. sclerotiorum* genes in SsMYRV4 infection were examined by GO enrichment analysis in biological processes, respectively, and the results were compared to the enriched GO terms of *S. sclerotiorum* genes that have similar codon usage patterns to mycoviruses. There were eight identical GO terms between the enriched GO terms of *S. sclerotiorum* genes that have similar codon usage pattern to mycoviruses with the GO enrichment results of upregulated genes for SsMYRV4 (Table 1 and Appendix A) involved in RNA processing, compound, RNA, and nucleic acid metabolic process; it also enrichened one GO term involved in transcription (GO:0006360) (Appendix A). There were no identical GO terms between the enriched GO terms of *S. sclerotiorum* genes that have similar codon usage pattern to mycoviruses with the GO enrichment results of downregulated genes for SsMYRV4 (Appendix A), suggesting that *S. sclerotiorum* genes with similar codon usage to mycoviruses were upregulated upon SsMYRV4 infection.

Lee et al., (2014) and Wang et al., (2016) reported transcriptional changes in *F. graminearum* infected by Fusarium graminearum virus 1 (FgV1), FgV2, FgV3, FgV4, and Fusarium graminearum hypovirus 1 (FgHV1) [44,45]. The upregulated and downregulated *F. graminearum* genes upon FgV1, FgV2, FgV3, FgV4, and FgHV1 infections were also examined by GO enrichment analysis in biological processes, respectively, and the results were also compared to the enriched GO terms of *F. graminearum* genes that have similar codon usage pattern to mycoviruses. There are no identical GO terms between the enriched GO terms of *F. graminearum* genes that have similar codon usage to mycoviruses with the GO enrichment results of differentially expressed genes for FgV3, FgV4, and FgHV1 (Appendix A); the same results appeared in the downregulated genes for FgV1 and FgV2 (Appendix A). However, there are 13 identical GO terms between the enriched GO result of *F. graminearum* genes that have similar codon usage pattern to mycoviruses and the GO enrichment result of the upregulated genes for FgV2, which almost covered all the enriched GO terms of upregulated genes for FgV2, and includes eight identical GO terms of *F. graminearum*, *S. sclerotiorum*, and *B. cinerea* (Table 1 and Appendix A), suggesting that *F. graminearum* genes with similar codon usage to mycoviruses were upregulated upon FgV2 infection. Although the GO enrichment result of upregulated genes for FgV1 share no identical GO term with the enriched GO terms of *F. graminearum* genes that have similar codon usage pattern to mycoviruses, it enriched one GO term involved in transcription (GO:0045944) (Appendix A), while the differentially expressed genes for FgV3, FgV4, and FgHV1 were not enriched in the GO term involved in transcription or RNA biosynthetic process, indicating that FgV1 can upregulate *F. graminearum* transcriptional genes with similar codon usage relative to mycoviruses.

## 3. Discussion

CUB is a common phenomenon and has been analyzed in viruses infecting plants, insects, animals, and fungi [17,19,20,32,33,34,46]. However, a systematic analysis of CUB has not been conducted in mycoviruses that infect plant pathogenic fungi. In this study, the codon usage patterns of 98 RNA mycoviruses from eight phytopathogenic fungi were investigated. In contrast to previous studies on mycoviral codon usage, our analysis was more detailed and systematic, and we also analyzed the relationship between mycoviruses and fungi at the single gene level.

ENc is a simple measure for estimating the overall CUB in whole viral genomes. Weak CUB was previously detected in RNA viruses, such as the Ebola virus [47], Zika virus [38], SARS-CoV-2 [48], and *Aspergillus*-infecting mycoviruses [32]. We also found that 97 RNA mycoviruses have high ENc values (>35), except for RsBPLV1 (28.28), suggesting that CUB in mycovirus is weak. This was consistent with *Aspergillus*-infecting mycoviruses and *Partitiviridae* mycoviruses [32,33]. A possible explanation is that weak viral CUB favors adaptation to the host using various preferred codons, promoting efficient viral replication and transcription by reducing the competition machinery for protein synthesis between viruses and hosts [21]. The ENc values of *R. solani* genes, *F. graminearum* genes, *S. sclerotiorum* genes, and *B. cinerea* genes were also calculated (Appendix A). The ENc values of 101, 819, 177, and 289 genes were less than 40 in *R. solani*, *F. graminearum*, *S. sclerotiorum*, and *B. cinerea*, respectively, and the codon usage patterns between these genes and mycoviruses show high difference. The fungal genes of *R. solani*, *F. graminearum*, *S. sclerotiorum*, and *B. cinerea* that have similar codon usage pattern to mycoviruses show weak CUB. High CUB is present in several viruses (for instance, Lymantria dispar nucleopolyhedrovirus and Orgyia pseudotsugata nucleopolyhedrovirus that are usually correlated with a high GC%) [20]. Indeed, RsBPLV1 has a high GC% (54.92%) and GC3s content (97.13%) and high CUB.

A previous study has reported that there is a moderate positive correlation of GC content between fungi and mycoviruses [34]. In our analysis, the GC and GC3s values of mycoviruses have a large span, 29.35% to 64.62% and 24.32% to 97.13%, respectively. The GC3s average contents of *R. solani* genes, *F. graminearum* genes, *S. sclerotiorum* genes, and *B. cinerea* genes are 50.80%, 53.17%, 40.02%, and 41.52%, respectively. By conducting a comparative analysis of GC3s contents between mycoviruses and their fungi hosts, we found that the more significant difference in GC3s contents between the mycoviruses and their fungi host genes, the greater the difference of codon usage pattern compared to host genes, and GC3s contents between mycoviruses and genes are similar, and their codon usage patterns are similar (Figure 6, Appendix A).

The relative dinucleotide abundance can also affect CUB in viruses [37]. Our analysis showed that the occurrence of dinucleotide was not randomized, and mycoviruses typically used CpA, GpA, and TpG (Figure 5 and Appendix A). The frequencies of CpA, GpA, and TpG were also in the top four in fungi hosts (*R. solani*, *F. graminearum*, *S. sclerotiorum*, and *B. cinerea*) (Appendix A). Due to the susceptibility of uracil in TpA to RNase [49] and low thermal stability [50], TpA content is restricted in many organisms. In mycoviruses, TpA was suppressed, and mycoviruses avoided using codons containing TpA, in *R. solani*, *F. graminearum*, *S. sclerotiorum*, and *B. cinerea*; TpA was also suppressed (). CpG dinucleotide is the foremost broadly and deeply studied dinucleotide in virus genome recently, and CpG is under-represented in many RNA viruses [36,51]. The frequency of CpG in mycoviruses and fungi hosts was not under-represented, but it fell within the normal frequency range (Appendix A). (−)ssRNA mycoviruses have the lowest mean CpG frequency (0.82 ± 0.31) compared with those of dsRNA (0.86 ± 0.18) and (+)ssRNA mycoviruses (0.99 ± 0.16). This finding is similar to the previous CpG analysis of RNA viruses [36]. However, the CpG values of mycoviruses are relatively higher in our analysis than in previously reported analyses.

As obligate intracellular parasites, mycoviruses are entirely dependent on the host cellular system to replicate [29]. Thus, the codon usage pattern of viruses may be likely affected by host. Such as poliovirus’s codon usage pattern is mostly coincident with that of its host [52]. In contrast, the hepatitis A virus has a codon usage pattern that is the opposite of that of its host [53]. Unlike completely identical or opposite patterns, some viruses have evolved a mixture of coincident and antagonistic codons, for instance, Citrus tristeza virus [18], Zika virus [25], and chikungunya virus [54]. Jitobaom et al. (2020) analyzed codon usage similarity between human RNA viruses and human genes, which demonstrated that the codon usage of cell-cycle-regulated genes are similar to that of human RNA viruses, indicating that human RNA viral genes may be efficiently expressed at the certain stages of the cell cycle [55]. In this study, we found that the fungal genes involved in nucleobase-containing compound metabolic/biosynthetic process, heterocycle biosynthetic process, RNA biosynthetic process, and transcription that have similar codon usage pattern to mycoviruses are significantly enriched. Chen et al. (2020) confirmed that when an exogenous gene has a CUB more similar to the host, it can impose minimum translational loads on the host cell, whereas the expression efficiency of the exogenous gene was improved [56]. Similar codon usage of mycoviruses and fungal genes involved in the transcription and RNA biosynthetic process can decrease the load on the host when mycoviruses replicate and transcribe in the host, and this can be beneficial to the self-replication and translation of mycoviruses in the host. This phenomenon could be an adaptative mechanism used by mycoviruses in resistance to fungi host immunity during long-term co-evolution.

Human genes that have similar codon usage to human immunodeficiency virus type 1, Zika virus, influenza A virus, and dengue virus serotype 2 may be upregulated in viral infections [55]. In our analysis, 9 and 13 GO terms of *S. sclerotiorum* and *F. graminearum* genes that have similar codon usage pattern to mycoviruses have been found to be identical to the GO terms of upregulated *S. sclerotiorum* and *F. graminearum* genes in SsMYRV4 and FgV2 infection, respectively; FgV1 infection can also upregulate *F. graminearum* transcriptional genes that with similar codon usage to mycoviruses, while there was no effect on the *F. graminearum* genes that have similar codon usage pattern to mycoviruses during FgV3, FgV4, and FgHV1 infections. This may be due to SsMYRV4, FgV1, and FgV2 are hypovirulence-associated mycoviruses [44,57], and FgV3, FgV4, and FgHV1 are latent mycoviruses [44,58].

The research on mycoviruses Fusarium oxysporum f. sp. dianthimycovirus 1, Heterobasidion partitivirus 13 strain an1, Phytophthora endornavirus 2 (PEV2), and PEV3 show that the high titer of mycoviruses can induce more severe symptoms in the host than mycoviruses with low titer [59,60,61]. Hypovirulence-associated mycoviruses can cause the hypovirulence of hosts and this may be due to hypovirulence-associated mycoviruses’ ability to upregulate host genes that have similar codon usage patterns to mycoviruses, and this may be beneficial in increasing their accumulation; while latent mycoviruses cannot upregulate the hosts genes that have similar codon usage, their accumulation may be at a low level, and they cannot cause a hypovirulent host.

There are still several analyses that need to be conducted on mycovirus codons usage. Some mycoviral genomes have more than one ORF. For example, members within the family Partitiviridae have two ORFs (encoded coat protein and RNA-dependent RNA polymerase). Generally, genes in the same plant virus have similar codon bias patterns, and they are slightly different [17]. However, the codon bias patterns of different genes in the same mycovirus are still unknown. Several mycoviruses are allocated to families that contain members infecting animals or plants, and it will be of great interest to analyze whether these mycoviruses show codon bias patterns similar to animal/plant-infecting members.

## 4. Materials and Methods

### 4.1. Sequence Data Acquisition

The fungal gene sequences data and mycoviral RP sequences were retrieved from the NCBI database based on the following criteria [62]: (i) The gene has a complete coding sequence (CDS); (ii) the length of the CDS is more than 300 nucleotides (nt); (iii) the CDS starts with ATG and ends with a stop codon; (iv) the CDS has no degenerate bases. Detailed information for all chosen mycoviruses was listed in Appendix A. The BioSamples of *Rhizoctonia solani*, *Fusarium graminearum*, *Sclerotinia sclerotiorum*, and *Botrytis cinerea* are SAMEA2810501, SAMEA3283145, SAMN05908049, and SAMN02953628, respectively.

The nucleotide compositional were calculated for the RP genes utilizing the CAIcal web application (http://genomes.urv.es/CAIcal) (assessed on 15 June 2020): (i) A%, C%, T%, G%, and GC%; (ii) GC3s% (the frequency of the third nucleotides G + C in the synonymous codons). The nucleotide composition of each mycovirus was detailed in Appendix A.

### 4.2. The Effective Number of Codons Analysis

ENc is presented by Wright at 1990 to quantify the synonymous codon usage bias, and it is independent of gene length and amino acid composition [35]. ENc values range from 20 (using only one synonymous codon) to 61 (all synonymous codons are used equally). A high ENc value indicates a low CUB, whereas a low ENc value signifies a high CUB. A value of 35 is a typical cutoff point [35,63]. Mycoviral ENc values were calculated using CodonW 1.4.4 software (http://codonw.sourceforge.net) (accessed on 15 June 2020). The ENc values of fungi host genes and mycoviruses were detailed in Appendix A.

### 4.3. ENc-GC3s Plot Analysis

An ENc-GC3s plot was drawn using ENc (the effective number of codons) values as the *Y*-axis and the GC3s values as the *X*-axis. The plot was used to examine the contribution of mutation and selection in forming mycoviral CUB. In the case of mycoviral ENc values that are lying on or just under the curve of the expected ENc values, this means that CUB is constrained by mutation pressure. Otherwise, natural selection is the main force in shaping the mycoviral CUB [35]. The expected ENc values for each GC3s were calculated using the following formula:$$\mathrm{ENc}\;=2+\;s\;+\frac{29}{s^{2}+{\left(1-s\right)}^{2}}$$
in which “*s*” stands for GC3s value [35].

### 4.4. Parity Rule 2 (PR2) Analysis

The PR2 plot shows the AT-bias (A3/(A3 + T3)) versus the GC-bias (G3/(G3 + C3)) at the third codon position of the four-codon amino acids: alanine, glycine, proline, threonine, valine, arginine (CGA, CGT, CGG, and CGC), leucine (CTA, CTT, CTG, and CTC), and serine (TCA, TCT, TCG, and TCC). The midpoint equaling 0.5 (A = T and G = C) in the PR2 plot indicates that mutation and selection pressure play the same role in shaping CUB; in the case where mycoviruses are lying on or near the midpoint, this means that CUB is constrained by mutation pressure. Otherwise, natural selection is the main force in shaping mycoviral CUB [64].

### 4.5. Neutrality Plot (GC12 vs. GC3) Analysis

The mycoviral GC values at the first (GC1), second (GC2), and third (GC3) positions of codons were calculated. Three stop codons (TAA, TAG, and TGA) and three isoleucine codons (ATT, ATC, and ATA) were eliminated from GC3 calculation. In addition, ATG and TGG were excluded in GC1, GC2, and GC3 analyses. GC12 stands for the average of GC1 and GC2. A neutrality plot (GC12 vs. GC3) was drawn using GC12 as the *Y*-axis and GC3 as the *X*-axis. The neutrality plot was used to quantify the relative contributions of mutation pressure and natural selection in shaping mycoviral CUB. Each dot represents an independent mycoviral RP gene. In a neutrality plot, when the regression coefficient is near 1, CUB is mainly affected by mutation pressure and is the dominant factor. In contrast, if the regression coefficient is near 0, natural selection plays a key role [65].

### 4.6. Relative Dinucleotide Abundance Analysis

The relative dinucleotide frequency was characterized as the observed frequency of a dinucleotide pair in a given sequence divided by the frequencies of the two mononucleotides that shape the dinucleotide pair in the same sequence. The relative dinucleotide frequency was calculated based on the following formula:$$\rho_{xy}=\frac{fxy}{fxfy}$$
in which “*fx*” and “*fy*” stand for the frequencies of nucleotides X and Y, respectively, and “*fxy*” stands for the observed dinucleotide XY value (calculated with DAMBE software). The XY dinucleotide was over-represented when *ρ_xy_* > 1.23 and under-represented when *ρ_xy_* < 0.78 [66]. The frequencies of 16 dinucleotides of fungi host genes and mycoviruses were detailed in Appendix A.

### 4.7. Principal Component Analysis of RSCUs

The relative synonymous codon usage (RSCU) value represents the observed codon frequency ratio to the expected codon frequency, given that all synonymous codons for any particular amino acid are used identically in a gene. RSCU accurately expresses the synonymous codon frequencies without the confounding factors of amino-acid composition and sequence length. The RSCU value was calculated following a previously described method; ATG, TGG, and TAA in addition to TAG and TGA, were eliminated from the analysis [67]. An RSCU value > 1.0, <1.0, and =1.0 indicates a positive, negative, and null CUB, respectively. Moreover, codons with an RSCU value > 1.6 were regarded as over-represented, whereas those with an RSCU value < 0.6 were considered under-represented [68].

The RSCU patterns of fungal single-gene and mycoviral RP genes were calculated (Appendix A). Principal component analysis (PCA) was used to reflect the similarities of codon usage patterns between mycoviruses and fungal genes. The RSCU pattern of each gene involves the distribution of 59 synonymous codons in 59 dimensions vector, and PCA can reduced the 59 values of each gene into the first two components: principal component 1 (PC1) and principal component 2 (PC2). Then, PC1 (*x*-axis) was plotted against PC2 (*y*-axis), and the RSCUs patterns of the fungal genes and mycoviruses were represented by PC1 and PC2 on the PCA graph. The PCAs of the RSCU graphs were analyzed using prcomp function in R.

### 4.8. Gene Ontology Enrichment Analysis

On the PCA graph of mycoviruses and fungal genes, the fungal genes that were in the same region of mycoviruses were taken as host genes that have similar RSCU patterns to mycoviruses. A great variation in RSCU patterns was observed among mycoviruses, and some mycoviruses were relatively concentrated together. To select the fungi host genes that have similar RSCUs to mycoviruses, the average PC1 and PC2 of the relatively concentrated mycoviruses were calculated; these mycoviruses were listed in Appendix A. Subsequently, the average PC1 and PC2 of those mycoviruses were set as the center of a circle with the radius calculated to be minimal and to cover those mycoviruses. The radius (*r*) from the circle center (*a*, *b*) was calculated as follows:$${\left(a\;–\;x\right)}^{2}+{\left(b\;–\;y\right)}^{2}=r^{2}$$in which *a* and *b* stand for the average PC1 and PC2 of mycoviruses in each host, and *x* and *y* stand for the PC1 and PC2 of the host genes.

The fungal genes with similar RSCUs to mycoviruses were analyzed for GO enrichment using the OmicShare tools, a free online platform for data analysis (https://www.omicshare.com/tools) (accessed on 7 October 2021); all fungal genes that were annotated to specific GO terms were used as the background. *p* < 0.01 was used as the threshold to evaluate significant GO terms in the biological process.

### 4.9. Correlation Analysis

Correlation analysis assessed possible relationships between variables. These analyses were carried out using the Pearson’s rank correlation test included in SPSS 18.0 for Windows. *p* < 0.01 and *p* < 0.05 indicate extremely significant and significant correlations, respectively.

## Figures and Tables

**Figure 1 ijms-23-07441-f001:**
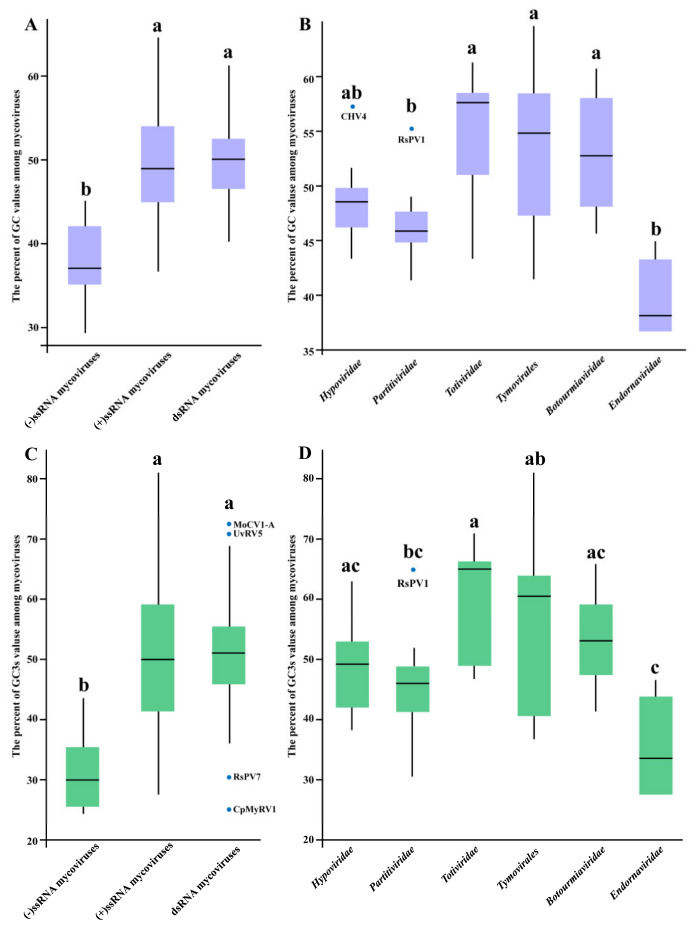
Comparative analysis of the GC and GC3s (the frequency of the third nucleotides G + C in the synonymous codons) contents in different mycoviruses. The *x*-axis represents (−)ssRNA, (+)ssRNA, and dsRNA mycoviruses in (**A**,**C**) and represents the *Hypoviridae*, *Partitiviridae*, *Totiviridae*, *Tymovirales*, *Botourmiaviridae*, and *Endornaviridae* mycoviruses in B and D. The *y*-axis represents the percent of GC values (**A**,**B**) and GC3s values (**C**,**D**) among (−)ssRNA, (+)ssRNA, and dsRNA mycoviruses (**A**,**C**), and *Hypoviridae*, *Partitiviridae*, *Totiviridae*, *Tymovirales*, *Botourmiaviridae*, and *Endornaviridae* mycoviruses (**B**,**D**). Statistical significance was determined by the LSD test. Different lowercase letters in the graph indicate statistical differences (*p* < 0.05), the statistical difference is not significant with same lowercase letter, and the different lowercase letters represent significant statistical differences.

**Figure 2 ijms-23-07441-f002:**
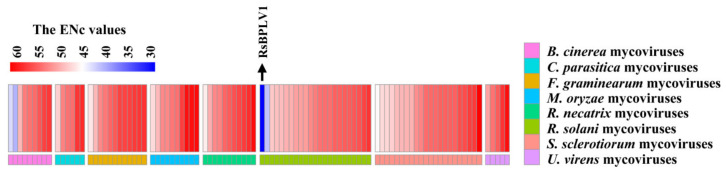
A heat map of the ENc (the effective number of codons) values of the selected mycoviruses, and it was drawn by pheatmap package in R. The legend was determined by the true ENc values of mycoviruses, the red represented the high value, the blue represented the low value, and the white represented the intermediate value among the 98 mycoviruses.

**Figure 3 ijms-23-07441-f003:**
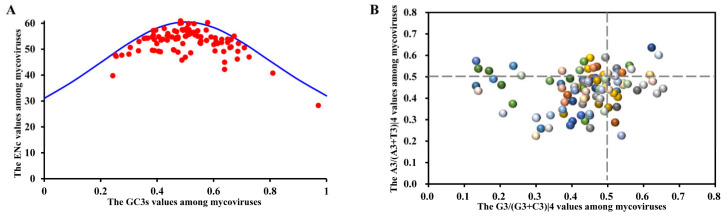
The ENc-GC3s plot (**A**) and Parity Rule 2 plot (**B**) of all mycoviruses. (**A**) The ENc-GC3s plot. The *x*-axis represents the GC3s (the frequency of the third nucleotides G + C in the synonymous codons) values of mycoviruses, and the *y*-axis represents the calculated ENc (the effective number of codons) values among mycoviruses. Each red point corresponds to one mycovirus, the solid blue line represents the expected ENc value when codon usage is only determined by GC3s composition. (**B**) The Parity Rule 2 plot. The *x*-axis represents the G3/(G3 + C3)|4 (the content of the third codon position of the four-codon amino acids) values, and the *y*-axis represents the A3/(A3 + T3)|4 values. Each three-dimensional ball represents a mycovirus, the color of the three-dimensional ball is automatically colored according to the corresponding data, and the intersection of gray dotted lines is the midpoint.

**Figure 4 ijms-23-07441-f004:**
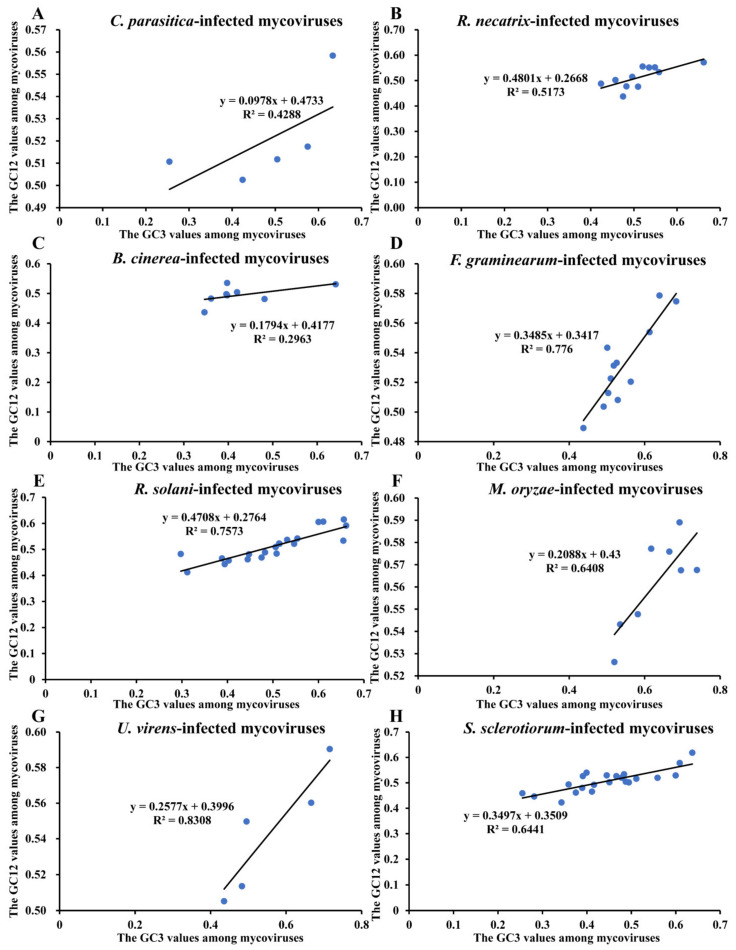
The neutrality plot analysis. The *x*-axis represents the GC3 (GC values at the third position of codons) values of mycoviruses, and the *y*-axis represents GC12 (the average of GC values at the first (GC1) and second (GC2) positions of codons) values of mycoviruses. The solid lines represent the correlation line, and the correlation curve equations are shown on the plot. Each blue dot represents one mycovirus. (**A**–**H**) represent the neutrality plot analysis of mycoviruses that infected *C. parasitica*, *R. necatrix*, *B. cinerea*, *F. graminearum*, *R. solani*, *M. oryzae*, *U. virens*, and *S. sclerotiorum*, respectively.

**Figure 5 ijms-23-07441-f005:**
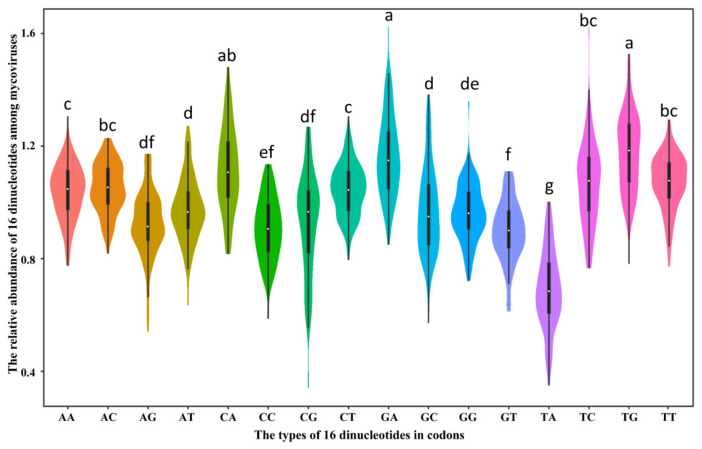
Violin plot illustrating the overall relative abundance of dinucleotides in mycoviruses. The *x*-axis represents the types of 16 dinucleotides in codons, and the *y*-axis represents the calculated relative abundance of dinucleotides among mycoviruses. Statistical significance was determined by LSD test. Different letters in the graph indicate statistical differences (*p* < 0.05), the statistical difference is not significant with same lowercase letter, and the different lowercase letter represents significant statistical differences.

**Figure 6 ijms-23-07441-f006:**
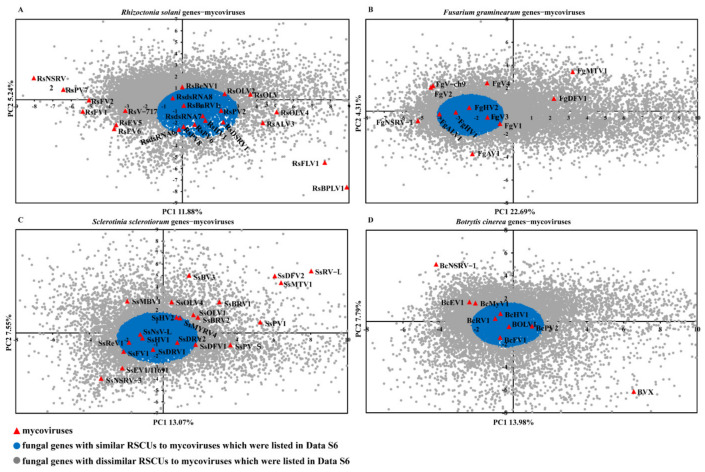
The PCA of the RSCUs among mycoviruses and fungal genes. The RSCU (the relative synonymous codon usage) values of fungal genes and mycoviruses were subjected to principal component analysis (PCA). The PCA was plotted by the first two axes (PC1 (*x*-axis) and PC2 (*y*-axis)). (**A**) *R. solani* genes and *R. solani* mycoviruses; (**B**) *F. graminearum* genes and *F. graminearum* mycoviruses; (**C**) *S. sclerotiorum* genes and *S. sclerotiorum* mycoviruses; (**D**) *B. cinerea* genes and *B. cinerea* mycoviruses. The red triangles represent the mycoviruses. The blue dots represent fungal genes that have similar RSCUs to mycoviruses, which were listed in Appendix A, and the gray dots represent the other fungal genes.

**Figure 7 ijms-23-07441-f007:**
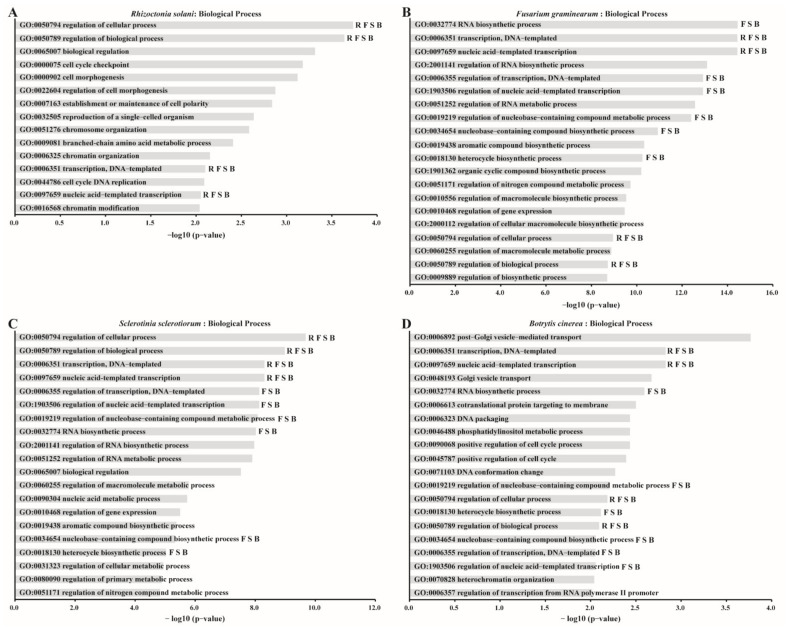
GO terms enrichment in the biological processes from fungal genes that have similar codon usage pattern to mycoviruses. The *Y*-axis shows the names of clusters of GO terms and the *X*-axis displays the −log10 (*p*-value) of corresponding GO term. GO terms with *p* < 0.01 were taken as a significant enrichment. (**A**–**D**) showed the top 20 of significant enriched GO terms from *R. solani* genes (**A**), *F. graminearum* genes (**B**), *S. sclerotiorum* genes (**C**), and *B. cinerea* genes (**D**), respectively. The alphabets indicate GO terms that which were co-enriched in different fungi; R (*R. solani*), F (*F. graminearum*), S (*S. sclerotiorum*), and B (*B. cinerea*).

**Table 1 ijms-23-07441-t001:** The identical GO terms in the biological processes between host genes with similar codon usage to mycoviruses and the upregulated host genes upon mycoviruses infections.

Mycovirus	GO Term	Description
SsMYRV4	GO:0006725	cellular aromatic compound metabolic process
	GO:0044260	cellular macromolecule metabolic process
	GO:0046483	heterocycle metabolic process
	GO:0090304	nucleic acid metabolic process
	GO:0006139	nucleobase-containing compound metabolic process
	GO:1901360	organic cyclic compound metabolic process
	GO:0016070	RNA metabolic process
	GO:0006396	RNA processing
FgV2	GO:0019438	aromatic compound biosynthetic process
	GO:0018130	heterocycle biosynthetic process
	GO:0097659	nucleic acid-templated transcription
	GO:0034654	nucleobase-containing compound biosynthetic process
	GO:1901362	organic cyclic compound biosynthetic process
	GO:2000112	regulation of cellular macromolecule biosynthetic process
	GO:0010556	regulation of macromolecule biosynthetic process
	GO:1903506	regulation of nucleic acid-templated transcription
	GO:0019219	regulation of nucleobase-containing compound metabolic process
	GO:2001141	regulation of RNA biosynthetic process
	GO:0051252	regulation of RNA metabolic process
	GO:0006355	regulation of transcription, DNA-templated
	GO:0032774	RNA biosynthetic process
	GO:0006351	transcription, DNA-templated

## Data Availability

The data presented in this study are openly available in Mendeley Data at http://doi.org/10.17632/tr8y8tgcyv.2.

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
