# Peer review of "Codon Usage Provides Insights into the Adaptive Evolution of Mycoviruses in Their Associated Fungi Host"

_ijms, 2022, doi:10.3390/ijms23137441_

Round 1

Reviewer 1 Report

Codon usage bias is a fundamental biology question that is not completely understood. Examining the codon usage in viruses is important to our understanding of their evolutionary pathway along with the hosts. Here, Wang et al performed bioinformatic analyses of the RNA mycoviruses genomes, from eight phytopathogenic fungi hosts. Authors lay out data from multiple types of analyses, yet their biological meanings were poorly explained. 

Major point:

1. Authors talk about co-evolution of mycoviruses with their fungi hosts, and analyzed large datasets, but never showed any phylogenetic relationships among these viruses or among their hosts.

2. Figure 1, it is interesting to see that (-)ssRNA mycoviruses have lower GC values in their genomes. Is this true for viruses that infect other type of hosts? And what is the potential reason for such a characteristic?

3. It is almost impossible to get the meaning of Fig. 2. Authors should do a better job to describe the data plotted.

4. Authors cite earlier work on using ENc, PR2, neutrality plot analyses to describe codon usage, and it is extremely difficult for readers that are less familiar with these analyses to understand how the relate the numbers to the biological meaning.

5. The dinucleotide analysis was performed by pooling the data from all mycoviral RP sequences from different viruses. Shouldn't the authors analyze each viral genome independently and then compare with the corresponding host genome?

6. Figure 7, surprising to not see translation coming up in the GO analysis, which is directly related to codon usage.

7. line 272-284, if codons are found in mycovirus genomes while the corresponding tRNAs are missing in the host, how are the genomes translated and how do these mycoviruses replicate?

8. line 359-363, isn't this the reason why one would use GC3s as a surrogate of codon usage pattern, and thus expected even without looking at the data?

Minor points:

line 105, define RSCU.

line 123, define ENc, what is effective number of codons?

line 178, what is GC12 value?

line 193, is CAT a typo?

Reviewer 2 Report

The authors of this study have examined the genome content, codon usage bias of mycoviruses that infect plant pathogenic fungi. Their insilico analysis showed that mycoviruses share codon usage pattern similar to that of some of the fungal genes. Authors of this study had concluded that mycoviral codon usage pattern is probably due to co-evolutionary changes between viruses and hosts. Experiments plan is good were decent number of sequences had been included in the study.

Author Response

Recruiting Reviewers

Round 2

Reviewer 1 Report

Authors have adequately addressed my points. Congrats.
